# LiNC: Lightweight Noise Correction via Adaptive Label Refinement

## Abstract

Medical imaging datasets often contain label noise due to factors such as inter-rater variability, annotation errors, and ambiguous cases, which may severely undermine the reliability and clinical effectiveness of machine learning models trained using those datasets. To address this challenge, we introduce Lightweight Noise Correction (LiNC), which is an intuitive and powerful approach that assigns a trainable trust parameter, $\alpha_i$, to each individual training sample. Initially initialized to fully trust the observed labels, these parameters adaptively shift trust towards model predictions through a gradient-based optimization process, effectively identifying and reducing the impact of noisy labels by correcting them. After this correction process, usual model training is carried out. Our method requires minimal computational overhead, making it practical for widespread adoption in cases where noise is suspected within a dataset. Extensive evaluations on ten medical imaging datasets from the MedMNISTv2 collection reveal significant improvements in classification accuracy and AUROC across various uniform label noise levels (ranging from 0% to 50%) and robust detection of mislabeled samples, underscoring LiNC's potential to improve noisy machine learning.

## 1 Introduction

Within healthcare, medical imaging is essential in supporting clinical tasks such as diagnosis, treatment planning, and disease monitoring. Recent advancements in deep learning have significantly improved medical image analysis by automating the detection and classification of various medical conditions. However, these advancements heavily depend on the availability of accurately labeled datasets. Machine learning models tend to severely degrade in performance when trained on noisy data. On the other hand, label noise is prevalent in healthcare datasets due to inconsistent annotations, ambiguous findings, and human errors during the annotation process. As a result, these kinds of inaccuracies can pose severe limitations to the reliability of machine learning models and may potentially have adverse effects on patient outcomes if not dealt with properly.

Current approaches to handling label noise predominantly include methods such as sample weighting, output regularization, non-self label correction, and self-label correction, as discussed in Related Work. While some of these methods may be effective at mitigating the effects of label noise, not all of them correct label noise and even when they do, they often may need a high computational load, tedious hyperparameter tuning, additional loss terms, clean validation sets, and may not even take the evolving training dynamics into account. This limits their use in practical clinical settings.

To overcome these limitations, we propose Lightweight Noise Correction (LiNC), a simple yet powerful methodology designed to improve accuracy, interpretability, and reliability in medical imaging classification tasks. LiNC introduces a differentiable, per-sample parameter, $\alpha_i$, which initially places full trust in the observed labels but dynamically shifts trust towards the model's predictions based on the learned confidence levels, which is automatically learned through the training process. This approach not only corrects noisy labels effectively but also provides direct insight into label reliability, through the analysis of the $\alpha_i$ parameters.

In summary, we introduce a new method, LiNC, that:

1. adds one extra trainable parameter per sample and only needs to update the parameters per-batch using manual gradient update (maintains similar time/space complexity);

2. performs label correction based on learned knowledge to improve model training and enhance test performance (does not require weighting or pruning valuable data);

3. does not require a clean validation set or meta-set for the method (may not necessarily be available in the health domain); and,

4. is lightweight, flexible, and can be easily incorporated into existing training or fine-tuning workflows.

## 2  RELATED WORK

The challenge of training deep neural networks in the presence of label noise has motivated several methods designed to either suppress noise or directly correct errors. Although sample weighting methods (Katharopoulos & Fleuret, 2019; Ren et al., 2019; Xu et al., 2021; Kong et al., 2021; Zhu et al., 2022; Zhou et al., 2023; Wu et al., 2023; Zhou et al., 2023; Wu & Li, 2024; Jain et al., 2024) aim to reduce the weight of certain samples based on difficulty (due to label noise, input noise, ambiguous samples, etc.), label modification can help more directly correct and improve the training trajectory.

As mentioned by Wang et al. (2022) in great detail, there are two main categories of label modification: output regularization, which works by adjusting the confidence levels of the targets, and label correction, which involves modifying the labels based on some information.

Output regularization aims to prevent models from becoming overly confident in their predictions with methods such as: label smoothing (Müller et al., 2020) and confidence penalty (Pereyra et al., 2017). Label smoothing introduces uncertainty into labels by replacing the hard one-hot target distribution with a soft, convex combination of the original label and a uniform distribution across all classes to help reduce overfitting and encourage better generalization, particularly in clean-label scenarios. Similarly, confidence penalty discourages low-entropy output distributions. Both of these methods help improve calibration and robustness on clean or mildly corrupted datasets. *However, they do not perform label correction, so they are not as useful in high-noise scenarios where incorrect labels need to be explicitly identified and corrected.*

To address this, label correction strategies have been developed to revise the labels based on more reliable estimates of the true label distribution. These approaches can be further divided into non-self and self label correction methods.

Non-self label correction relies on external models - called teacher networks - to generate soft labels that are used as improved supervision for the student network. An example of non-self learning correction is knowledge distillation (Hinton et al., 2015), where a pretrained or concurrently trained teacher model provides probabilistic targets for the student model. This uses knowledge from a potentially more stable or better-calibrated model to hopefully remove noise in the observed labels and improve training dynamics. *However, non-self label correction approaches inherently require significant overhead and are susceptible to performance degradation if the teacher model is biased or miscalibrated.*

Self label correction approaches aim to refine labels using the model's own predictions, removing the need for additional models and improving chances for scalability and practical deployment. These methods are simpler and more end-to-end compatible. One such approach proposed by Lee et al. (2013) replaces the observed label with the model's current prediction by using the highest-probability class. In some cases (Vyas et al., 2020), soft-labels may make training unstable. *However, this method can suffer from confirmation bias, especially in early training phases when model predictions are still unstable and inaccurate.*

In some of these methods, incorrect predictions may be reinforced in self label correction methods, causing a feedback loop that worsens noise. To mitigate this, bootstrapping methods (Reed et al., 2014) blend the observed label with the model's current predictions using a convex combination. This interpolation is controlled by a fixed parameter that determines the trust in the model's predictions. These methods offer a trade-off between retaining potentially noisy labels and incorporating

corrections by the model. *However, the fixed nature of the parameter does not take the evolving trustworthiness of the model's predictions over the course of training into consideration.*

Other self label correction strategies attempt to address this by introducing stage-wise learning. Tanaka et al. (2018); Yuan et al. (2019); Lu et al. (2023) split training into stages where the model first learns from the observed, potentially noisy labels and later on, the model's predictions are fully trusted and used as labels. These models effectively deal with avoiding the need for a fixed trust parameter. *However, they still suffer from significant overhead.*

Most of these methods require additional models, hyperparameter tuning, manually chosen scores, trust in a potentially untrustworthy model or potentially unstable stage of training, and/or a higher computational load, making these methods less scalable and harder to deploy in real-world scenarios, especially in high-stakes domains like healthcare and medical imaging.

Therefore, in our method, we introduce a single extra differentiable parameter for each training sample that will use concepts from self label correction to either give more weight to the observed label or the model prediction after a few warmup epochs without requiring much additional overhead. Then, the method uses Otsu thresholding Otsu et al. (1975) on these smooth/convex parameters to separate samples into ones that will stick to their original label and ones that will switch to the label provided by the model. Unlike prior work, our method does not rely on extensive tuning, auxiliary models, or discrete training phases, and can be seamlessly integrated into standard training loops after a brief warmup phase, a brief correction phase, and then a final training phase offering a scalable and model-agnostic solution to label noise correction. In addition, our method also has potential to be very strong with pre-trained models when there is a need for fine-tuning rather than training from scratch for the best results.

## 3 METHOD

In this section, we provide a description of our method along with our overall training procedure. Let us consider the following supervised multi-class classification problem.

Let $\mathcal{D} = \{(x_i, \widetilde{y}_i)\}_{i=1}^N$ be the training set, where $(x_i, \widetilde{y}_i)$ are the inputs and potentially noisy, observed targets. In our case, $x_i \in \mathcal{X}$, where $\mathcal{X}$ is the input space of images and $\widetilde{y}_i \in \mathcal{Y} = \{0, ..., c-1\}$, is the output space with $c \in \mathbb{N}$ such that $c \geq 2$ is the number of classes. We also have the parameters $\alpha_i$ for $i \in \{1, ..., N\}$, i.e. one for each training sample. Let $f_\theta : \mathcal{X} \to \mathcal{Y}$ be the neural network model and $\theta$ be its parameters.

During training, our method optimizes both the network parameters and the $\alpha_i$ parameters. The training process is detailed in Algorithm 1.

The core mechanism underlying our noise correction framework is the interpolation of training targets using a per-sample trust parameter $\alpha_i \in [0, 1]$, which dynamically adjusts the degree of trust placed on the observed label versus the model's own prediction:

$$\widetilde{y}_i = (1 - \alpha_i) \cdot s_i + \alpha_i \cdot \widetilde{y}_i$$

When $\alpha_i = 1$: the model fully trusts the observed label $\widetilde{y}_i$.

When $\alpha_i = 0$: the model completely trusts its own prediction $s_i$.

Here, $\widetilde{y}_i$ represents the original observed label, and $s_i$ denotes the model's predicted probability distribution for a given sample. By treating $\alpha_i$ as a learnable trust coefficient that varies across samples and over time, we enable the model to self-modulate how much it sticks to the observed label versus its internal belief. This dynamic trust mechanism becomes useful in the presence of label noise when the model is given flexibility to assign more importance to its own prediction if it detects conflicts with the provided label.

During each training iteration, the $\alpha_i$ values are updated through a manual gradient descent step (as shown in Line 19 of Algorithm 1), with the objective of refining label reliability. We enforce a discrete decision and maintain stability during the training process (avoiding meaningless $\alpha_i$ values), after a

---

**Algorithm 1** Training with LiNC

---

1: **Inputs:** training data $\mathcal{D}$, classifier model $f_\theta$, trust parameters $\alpha_i$, number of warmup epochs $w$,
2:          optimizer for all $\alpha_i$ and $\theta$, learning rate $\alpha_{lr}$, weight decay $\alpha_{wd}$
3: **Output:** classifier model trained using LiNC
4: **for** epoch $\leftarrow 1$ **to** $n$ **do**
5:   **for** batch $(x, \widetilde{y}) \leftarrow$ dataloader($\mathcal{D}$) **do**
6:     set requires_grad to True on $\theta$ and $\{\alpha_j\}$ for current batch $j \subset \{1, ..., N\}$
7:     let $x_j = x$ and $\widetilde{y}_j = \widetilde{y}$
8:     $s_j = \texttt{softmax}(f_\theta(x_j))$
9:     **if** epoch $< w$ **then**
10:       $\widetilde{y}_j = (1 - \{\alpha_j\}) * s_j + (\{\alpha_j\}) * \widetilde{y}_j$
11:     **end if**
12:     **if** epoch $== w$ **then**
13:       $\widetilde{y}_j = s_j$, if $\{\alpha_j\} < \texttt{otsu}(\alpha_i)$
14:     **end if**
15:     calculate train loss $\mathcal{L} = \ell(f_\theta(x), \widetilde{y})$
16:     $\mathcal{L}$.backward()
17:     optimizer.step() to update $\theta$
18:     **if** epoch $< $ w **then**
19:       $\{\alpha_j\} = \{\alpha_j\} - \alpha_{lr} * \nabla_{\{\alpha_j\}}\mathcal{L}$
20:       $\{\alpha_j\} = \{0$ if $\alpha_j < 0$ and $1$ if $\alpha_j > 1\}$
21:     **end if**
22:     optimizer.zero_grad()
23:   **end for**
24: **end for**
25: **return** $f_\theta(x)$

---

few warmup epochs. We project the updated $\alpha_i$ values back to $[0, 1]$ in Line 20. Noise correction happens once in Line 13 after the warmup epochs.

Also, the behavior of the $\alpha_i$ parameters evolves throughout training. Samples with noisy or inconsistent labels tend to rapidly shift their $\alpha_i$ values towards 0, whereas clean samples generally stabilize with $\alpha_i$ values near 1. This separation implicitly segments the dataset into reliable and unreliable subsets, without requiring any prior knowledge of which samples are mislabeled. Furthermore, because the $\alpha_i$ updates are localized and differentiable, they can be used easily in any model training with minimal effort.

**Why Otsu on $\alpha$?**   We threshold $\alpha_i$ using Otsu's method to obtain a data-adaptive, parameter-free split between trusting the observed labels and trusting the model prediction. Otsu exploits bimodality that naturally emerges as clean samples keep $\alpha_i \approx 1$ while mislabels move toward 0. In addition, Otsu is much faster than most alternatives, differentiable, and introduces no extra hyperparameters.

## 4   EXPERIMENTAL SETUP

Our experiments are conducted on ten 2D datasets with different medical imaging modalities focusing on multi-class classification tasks: PathMNIST (Kather et al., 2019), DermaMNIST (Tschandl et al., 2018; Codella et al., 2019), OCTMNIST (Kermany et al., 2018), PneumoniaMNIST (Kermany et al., 2018), BreastMNIST (Al-Dhabyani et al., 2020), BloodMNIST (Acevedo et al., 2020), TissueMNIST (Ljosa et al., 2012), OrganAMNIST (Bilic et al., 2023; Xu et al., 2019), OrganCMNIST (Bilic et al., 2023; Xu et al., 2019), and OrganSMNIST (Bilic et al., 2023; Xu et al., 2019), from the MedMNISTv2 collection (Yang et al., 2021; 2023; Doerrich et al., 2024). All the images are of size 224×224 and we do not use any augmentations.

Using ImageNet-21K pretraining, we trained on 224×224 MedMNISTv2 inputs for 100 epochs with a batch size of 128, learning rate of $1e - 4$, and no weight decay. We used an Adam optimizer (Kingma, 2014) and a multi-step learning rate scheduler was applied, decaying the rate by a factor of 0.1 at epochs 50 and 75, following the setup in Yang et al. (2023).

We used a warmup period of 10 epochs to ensure that the model briefly learns about the mislabels in the data before using LiNC. We initially set the $\alpha_i$ parameters all to 1 (fully trust the observed labels), with learning rate $\alpha_{lr} = 1$ and weight decay $\alpha_{wd} = 1e-3$ for manual gradient descent. Our method is not too sensitive to this learning rate choice as the learning rate needs to be large enough to push the $\alpha_i$ for mislabels to 0 since generally, the $\alpha_i$ for correct labels do not change much at all. We inject various levels of uniform noise (0%, 10%, 20%, 30%, 40%, 50%) to study the effectiveness of LiNC. We analyze the classification accuracy and AUROC with and without LiNC and study its ability to identify mislabels compared to several baselines.

## 5 RESULTS

| Method | AUROC | |
|--------|---------|----------|
| | epoch 2 | epoch 10 |
| CNLCU-S | 0.6688 | 0.6829 |
| GraND | 0.6109 | 0.5785 |
| Data-IQ | 0.5275 | 0.6763 |
| DataMaps | 0.5000 | 0.4084 |
| EL2N | 0.6688 | 0.5606 |
| AUM | 0.6364 | 0.6794 |
| $\alpha_i$ | **0.7923** | **0.8012** |

Table 1: We report the AUROC scores of various methods in being able to identify mislabels when compared to the $\alpha_i$ parameters from our method after the warmup epochs on the BreastMNIST dataset with 50% uniform label noise.

In Table 1, we compare the $\alpha_i$ parameters from our method to several other baseline methods that have been effective in identifying noisy or otherwise "difficult" data: AUM (Pleiss et al., 2020), DataMaps (Swayamdipta et al., 2020), Data-IQ (Seedat et al., 2022), EL2N (Paul et al., 2021), GraND (Paul et al., 2021), and CNLCU-S (Xia et al., 2021). Note that most of these methods also require quite a bit more work to calculate and/or maintain whereas the $\alpha_i$ parameters from our method are simply maintained and updated during the training process.

Our method achieves the highest AUCROC score of 0.7923 after epoch 2 and 0.8012 after epoch 10, substantially outperforming all baselines. This result indicates that the learned $\alpha_i$ values are highly predictive of label correctness, providing strong evidence that our adaptive label correction mechanism is effective at isolating noisy labels early in training (just after 10 warmup epochs). These results highlight the practical benefits of our proposed method, which identifies trustworthiness through the $\alpha_i$ parameters without requiring auxiliary models or extensive post-processing. With this, we have shown that label correction can be formulated as a differentiable, per-sample trust mechanism which effectively learns to identify noisy supervision from within the training loop itself.

We observe that the per-sample trust parameters $\alpha_i$, exhibit a highly discriminative relationship between clean and noisy labels. Specifically, values of $\alpha_i$ are driven toward 1 for samples where the observed label is likely to be correct and values approach 0 for samples where the model prediction is deemed more reliable, indicating a lower degree of trust in the observed label. This happens consistently even under the use of a relatively large learning rate of 1 and weight decay of $1e-3$ for the $\alpha_i$ parameters. Despite the aggressive gradient updates, the optimization process maintains stability and convergence toward meaningful values that reflect the underlying quality of each label.

Therefore, our sample-wise label correction mechanism can dynamically infer which instances are mislabeled and adjust supervision accordingly. Even under minimal constraints and with highly localized updates, per-sample trust parameters can meaningfully disentangle noisy and clean samples and improve robustness to noise during training.

Table 2 shows the accuracy results of running the model with and without LiNC across ten 2D medical imaging datasets, of varying modalities, from the MedMNISTv2 (Yang et al., 2021; 2023; Doerrich et al., 2024) collection under varying degrees of uniform label noise (0%-50%). For each dataset, we report the baseline accuracy achieved without LiNC, and the accuracy improvements obtained

| | Acc. (%) | | | | | |
|---|---|---|---|---|---|---|
| **Dataset** | **0% Noise** | **10% Noise** | **20% Noise** | **30% Noise** | **40% Noise** | **50% Noise** |
| PathMNIST | 95.64 | 94.96 ↑ 0.48 | 94.70 ↑ 0.20 | 94.19 ↑ 0.41 | 93.27 ↑ 0.18 | 92.82 ↑ 0.92 |
| DermaMNIST | 73.70 | 72.01 ↑ 2.37 | 72.60 ↑ 1.98 | 70.87 ↑ 2.63 | 70.50 ↑ 1.37 | 69.87 ↑ 1.83 |
| OCTMNIST | 75.29 | 70.69 ↓ 0.15 | 69.84 ↑ 1.22 | 72.54 ↓ 3.10 | 68.08 ↓ 2.73 | 65.06 ↓ 0.39 |
| PneumoniaMNIST | 88.13 | 87.57 ↑ 3.10 | 87.14 ↑ 1.77 | 85.83 ↑ 3.05 | 86.03 ↑ 2.70 | 86.99 ↑ 0.73 |
| BreastMNIST | 83.26 | 84.26 ↑ 1.17 | 84.65 ↑ 0.39 | 80.92 ↓ 0.40 | 72.82 ↑ 9.66 | 69.87 ↑ 8.26 |
| BloodMNIST | 98.04 | 97.32 ↑ 0.78 | 96.21 ↑ 1.33 | 95.48 ↑ 1.60 | 94.15 ↑ 1.90 | 92.78 ↑ 1.78 |
| TissueMNIST | 67.16 | 65.95 ↓ 1.42 | 64.18 ↓ 1.03 | 63.31 ↓ 1.04 | 61.83 ↓ 0.76 | 60.01 ↓ 0.12 |
| OrganAMNIST | 94.37 | 92.78 ↑ 0.77 | 91.77 ↑ 0.60 | 90.64 ↑ 1.08 | 90.49 ↑ 0.66 | 89.27 ↑ 0.24 |
| OrganCMNIST | 87.44 | 85.12 ↑ 2.20 | 84.27 ↑ 2.49 | 82.22 ↑ 2.64 | 80.76 ↑ 2.55 | 78.61 ↑ 2.97 |
| OrganSMNIST | 77.49 | 75.43 ↑ 1.61 | 73.65 ↑ 1.92 | 71.52 ↑ 2.92 | 70.21 ↑ 2.97 | 65.10 ↑ 3.96 |

Table 2: Accuracy on ten 2D datasets from MedMNISTv2 with different levels of uniform noise.

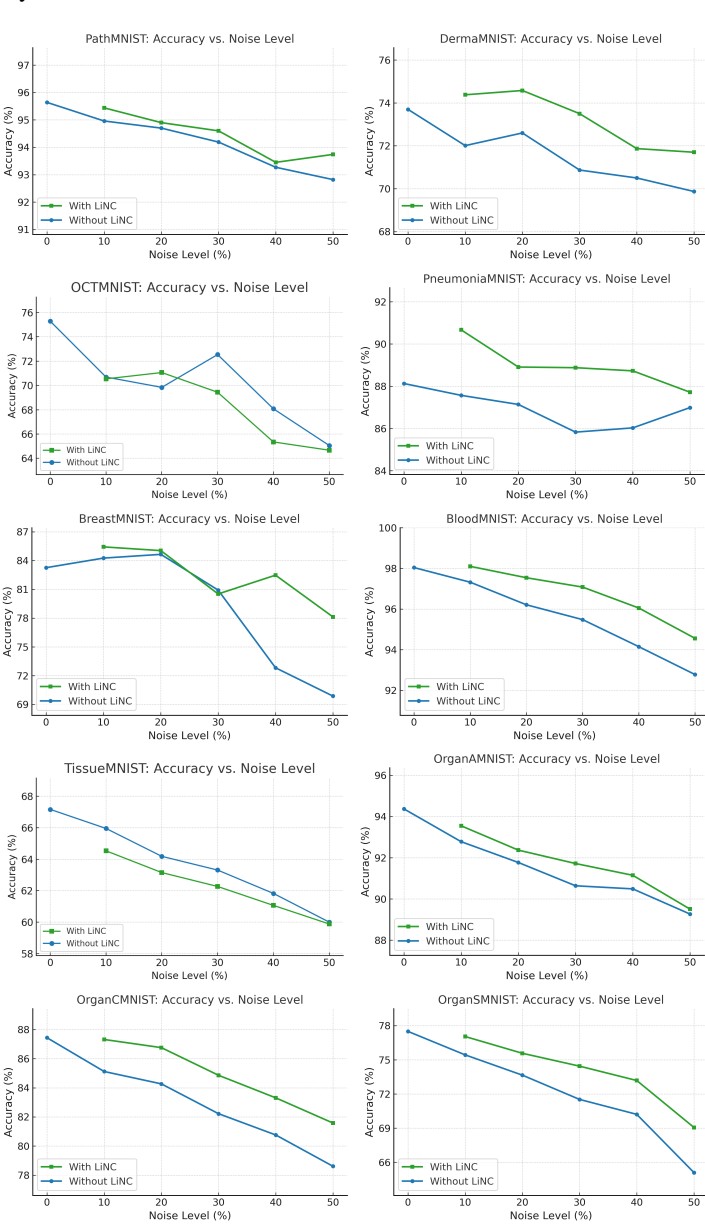

Figure 1: Accuracy with and without LiNC on ten 2D datasets from MedMNISTv2.

| | AUROC | | | | | |
|---|---|---|---|---|---|---|
| **Dataset** | **0% Noise** | **10% Noise** | **20% Noise** | **30% Noise** | **40% Noise** | **50% Noise** |
| PathMNIST | 0.9968 | 0.9934 ↑ 0.0033 | 0.9912 ↑ 0.0050 | 0.9898 ↑ 0.0053 | 0.9881 ↑ 0.0052 | 0.9862 ↑ 0.0057 |
| DermaMNIST | 0.8936 | 0.8630 ↓ 0.0063 | 0.8536 ↑ 0.0065 | 0.8338 ↑ 0.0278 | 0.8004 ↑ 0.0489 | 0.7845 ↑ 0.0395 |
| OCTMNIST | 0.9817 | 0.9691 ↑ 0.0139 | 0.9626 ↑ 0.0127 | 0.9580 ↑ 0.0114 | 0.9569 ↓ 0.0019 | 0.9380 ↑ 0.0089 |
| PneumoniaMNIST | 0.9646 | 0.9706 ↑ 0.0092 | 0.9659 ↑ 0.0063 | 0.9666 ↑ 0.0056 | 0.9516 ↑ 0.0164 | 0.9441 ↑ 0.0146 |
| BreastMNIST | 0.8678 | 0.8680 ↓ 0.0046 | 0.8594 ↑ 0.0001 | 0.8446 ↓ 0.0173 | 0.7878 ↑ 0.0370 | 0.7621 ↓ 0.0023 |
| BloodMNIST | 0.9984 | 0.9976 ↑ 0.0012 | 0.9967 ↑ 0.0016 | 0.9954 ↑ 0.0019 | 0.9933 ↑ 0.0023 | 0.9908 ↑ 0.0020 |
| TissueMNIST | 0.9074 | 0.9035 ↓ 0.0363 | 0.8942 ↓ 0.0399 | 0.8877 ↓ 0.0426 | 0.8777 ↓ 0.0486 | 0.8623 ↓ 0.0403 |
| OrganAMNIST | 0.9865 | 0.9940 ↑ 0.0015 | 0.9929 ↑ 0.0018 | 0.9926 ↑ 0.0013 | 0.9912 ↑ 0.0006 | 0.9889 ↑ 0.0011 |
| OrganCMNIST | 0.9928 | 0.9833 ↑ 0.0032 | 0.9819 ↑ 0.0038 | 0.9783 ↑ 0.0044 | 0.9756 ↑ 0.0037 | 0.9723 ↑ 0.0043 |
| OrganSMNIST | 0.9670 | 0.9635 ↑ 0.0049 | 0.9564 ↑ 0.0070 | 0.9529 ↑ 0.0083 | 0.9496 ↑ 0.0087 | 0.9387 ↑ 0.0092 |

Table 3: AUROC on ten 2D datasets from MedMNISTv2 with different levels of uniform noise.

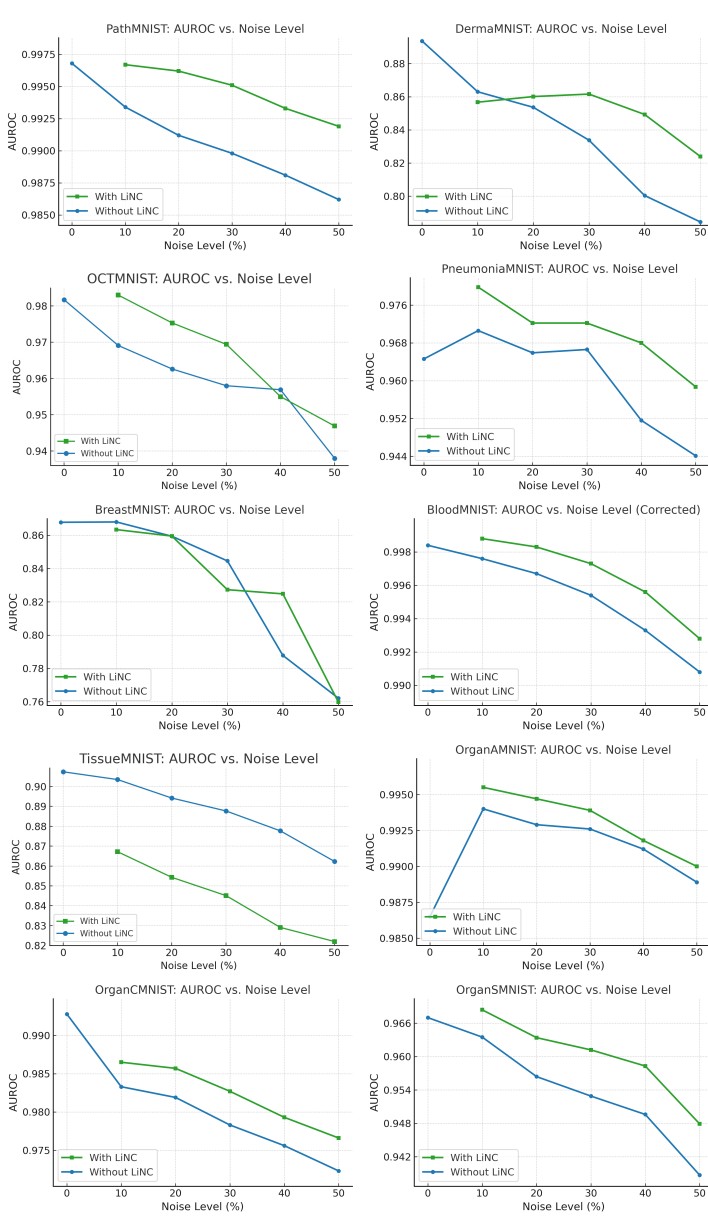

Figure 2: AUROC with and without LiNC on ten 2D datasets from MedMNISTv2.

when applying LiNC. Overall, the results consistently demonstrate that LiNC robustly improves classification accuracy across all datasets and noise levels examined. Specifically, the method

significantly mitigates the negative impact of label noise, with accuracy improvements becoming more pronounced as the noise level increases. This finding underscores LiNC's inherent capability of identifying mislabels and dynamically enabling the network to rely more strongly on its own progressively confident predictions rather than noisy training labels. It is especially noteworthy that in almost all cases where there is injected noise, the improvement in accuracy with LiNC is similar to the performance without LiNC when there is no injected noise. Figure 1 is a visual representation of Table 2. You can see that in almost all cases (except with OCTMNIST and TissueMNIST, which seem to be the hardest tasks on the MedMNISTv2 dataset), there is a significant increase in accuracy after correcting noise with LiNC.

Table 3 reports AUROC scores across the datasets. We see that large-scale or relatively easier datasets remain highly robust, retaining AUROC scores above 0.98 even at 50% noise, with only marginal fluctuations. More noise-sensitive datasets with AUROC scores less than 0.80 show degradations as noise increases, with TissueMNIST dropping from 0.9074 at 0% noise to 0.8623 at 50% while using LiNC. The remaining datasets show small but consistent improvements. It is also important to note that AUROC does not uniformly decrease with increasing noise. Figure 2 is a visual representation of Table 3.

## 6 DISCUSSION

Our results demonstrate the effectiveness of LiNC in mitigating label noise across various medical imaging datasets and different noise levels. The method consistently improved accuracy, indicating its robustness and adaptability in diverse medical imaging modalities, from dermatology to pathology. Notably, LiNC exhibited remarkable performance improvements even under extreme noise conditions (up to 50%), demonstrating its strength in accurately identifying and correcting mislabeled samples, which is crucial in high-stakes healthcare scenarios.

LiNC achieves these benefits with minimal computational overhead. By introducing only a single additional trainable parameter per sample, which dynamically adjusts based on evolving training dynamics, LiNC provides a scalable solution applicable to real-world clinical settings where computational resources and model interpretability are critical considerations. Model developers can use the $\alpha_i$ parameters as an easy way to perform human-in-the-loop audits of health data. Furthermore, LiNC's simplicity and flexibility mean it can be integrated into existing training workflows, including scenarios involving fine-tuning pretrained models or foundation models.

The adaptive nature of the parameters $\alpha_i$ allows LiNC to respond dynamically to the training progress. Our analysis of $\alpha_i$ indicates its effectiveness in differentiating between noisy and clean labels, as evidenced by consistently high AUCROC scores across different noise levels. This adaptability not only enhances accuracy but also provides interpretability into label reliability, offering valuable insights into dataset quality and annotation trustworthiness.

One potential limitation of LiNC is its reliance on self label correction. There may be cases where particularly challenging datasets or poorly initialized models could affect early-stage predictions, potentially impacting corrections. Future work could explore adaptive warmup strategies or more complex initializations to further enhance reliability.

Since LiNC significantly reduces the negative impact of noisy labels, we aim to investigate hybrid approaches that combine LiNC with active learning and human-in-the-loop verification to study dataset quality and maximize clinical applicability.

## 7 CONCLUSION

We presented LiNC, a lightweight, adaptive, and effective approach to address the issue of label noise in medical imaging datasets. By dynamically adjusting the trust placed in observed labels versus model predictions for each training sample, LiNC robustly improves classification accuracy, even in scenarios with substantial label noise. Our extensive experiments across diverse medical datasets demonstrate that LiNC not only enhances accuracy significantly but also provides interpretable insights into dataset reliability. Given its computational efficiency, ease of implementation, and proven effectiveness, LiNC represents a practical and powerful tool for improving the reliability of machine learning models in clinical settings by fixing noisy labels.

Future work can explore whether pseudo-labels, in the context of self-supervised or semi-supervised learning, can be fixed using LiNC. Additionally, LiNC may be useful for active learning techniques to identify which samples should be labeled (or relabeled) to most benefit the model.

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
