# OpenReview forum: "LiNC: Lightweight Noise Correction via Adaptive Label Refinement"
_ICLR.cc/2026/Conference — ICLR 2026 Conference Withdrawn Submission_

### Official Review · Reviewer_hJsW · 2025-10-16

**Soundness:** 1
**Presentation:** 1
**Contribution:** 1
**Rating:** 0
**Confidence:** 4

**Summary:**

The paper suggests a method for network training based on noisy labels. The technique gradually 'corrects' the labels by shifting them from the original label to the predicted class.

**Strengths:**

The paper addresses an important problem of handling a training set with unreliable labels.

**Weaknesses:**

The paper is not well written. The method is not well motivated and not clearly presented. An algorithm box (that is not well written) is not a replacement for a clear presentation and an algorithm motivation. Although network training based on noisy labels is a well-studied method, there is no comparison to previous works.

**Questions:**

What was the noise model you used in the experiments?

---

### Official Review · Reviewer_6wrZ · 2025-10-28

**Soundness:** 2
**Presentation:** 2
**Contribution:** 2
**Rating:** 4
**Confidence:** 4

**Summary:**

This paper introduced a noise correction technique in mechical image classification. It leveraged a trainable parameter for each data to reduce the negative impact of samples with higher noise level. The experimental results demonstrate that the methods can sucessfully get performance improvement on ten medical datasets.

**Strengths:**

1. Easy-to-plug: The methods can be directly applied in any classification framework with a plug-and-play manner. The core idea is direct and simple.

2. Wide range of evaluation: The authors tested the results on ten medical datasets with many noise levels and two metrics. The benchmark is comprehensive.

3. Motivation: The authors regarded alpha as "confidence", providing a novel perspective to the noise removal task.

**Weaknesses:**

1. The novelty is limited. Although authors provided a new perspective, the core part of the methods is still linear interpolation, which is highly similar to early bootstrapping. The paper lacks clear theoretical or empirical evidence that distinguishes it from these classic methods.

2. The claimed "differentiable" Otsu is confusing. The standard Otsu in the text is a discrete threshold selection rather than a differentiable operator.

3. Unprofessional result presentation and poor figure readability. Results in Table 1 reports AUROC only at fixed epochs(2/10), rather than following the standard ML protocol of reporting test performance from the best validation checkpoint. Furthermore, Figure 1 and 2 are less informative.

4. No key abliations. There is no systematic ablation on whether α continues to be updated, or alternative thresholding strategies (Otsu vs. others). Also there is no other comparison against the other baselines on the other datasets.

5. The noise setting is limited. Onlyl summetric uniform label noise is tested. In medical analysis, class-dependent or instance-dependent noise is more common. There should be comparison against other baselines such as Co-teaching [1], DivideMix [2], etc.

[1] Han, B., Yao, Q., Yu, X., Niu, G., Xu, M., Hu, W., ... & Sugiyama, M. (2018). Co-teaching: Robust training of deep neural networks with extremely noisy labels. Advances in neural information processing systems, 31.

[2] Li, J., Socher, R., & Hoi, S. C. (2020). Dividemix: Learning with noisy labels as semi-supervised learning. arXiv preprint arXiv:2002.07394.

**Questions:**

1. What is the core differences between LiNC and bootstrapping, label smoothing, and Mixup?

2. What is the evidence of "Otsu is differentiable"?

3. How would the performance go if conducting multi-round labeling and training?

4. How about the robustness of the model towards hyperparameters including learning rate, warmup epochs, threshold strategies, etc?

5. Why there is only one noise setting in Table 1?

6. Stronger baselines are suggested to be involved.

---

### Official Review · Reviewer_6jc6 · 2025-11-03

**Soundness:** 2
**Presentation:** 3
**Contribution:** 2
**Rating:** 2
**Confidence:** 4

**Summary:**

This paper tackles label noise in medical imaging through a straightforward pre-training approach called Lightweight Noise Correction (LINC). The idea is to identify and correct mislabeled samples before standard training begins, rather than trying to handle noise during training itself.
The core mechanism is simple.
1. The method assigns each training sample a learnable "trust parameter" $\alpha_i \in [0, 1]$, initially set to 1 (full trust in the label). During a short warmup phase, these parameters are updated via gradient descent alongside the model weights. The intuition is that clean labels produce low loss and small gradients, keeping $\alpha_i$ near 1, while noisy labels generate high loss and conflicting gradients that push $\alpha_i$ toward 0.
2. After warmup, the authors assume the $\alpha_i$ values have separated into a bimodal distribution—one mode for clean samples, another for noisy ones.
3.They use Otsu's thresholding to automatically find the split point between "trusted" and "suspicious" samples. The suspicious samples then have their labels corrected (typically replaced with the model's current prediction), and the model is retrained from scratch on this cleaned dataset.

The validation is performed on ten MedMNISTv2 datasets across various noise levels (0-50%). The results show that LINC effectively identifies mislabeled samples and improves both classification accuracy and AUROC compared to baselines that either ignore noise or use existing noise-handling techniques.

**Strengths:**

This paper's primary strength lies in its simplicity and practical utility for a common and difficult problem.
1. Originality and Simplicity: The paper introduces LiNC, an "intuitive and powerful approach"
 a. that boils the complex problem of noise correction down to a single, trainable "trust parameter" $\alpha_i$ assigned to each sample
 b. This design is conceptually simple, "lightweight"
c. and easy to grasp.
2. Practicality and Ease of Use: The method is significant because it is designed for real-world use. It is "lightweight, flexible, and can be easily incorporated into existing training... workflows", acting as a "plugin" module that refines the dataset.
3. It requires "minimal computational overhead", as it only "adds one extra trainable parameter per sample".
4. Most importantly, it "does not require a clean validation set or meta-set", which is a major practical advantage, especially in domains like healthcare where such clean data is often unavailable.

**Weaknesses:**

I have serious concerns about both the clarity of the methodology and the experimental validation. These issues make it difficult to assess whether the approach actually works as intended or offers any advantage over existing methods.
1. The most troubling problem is a fundamental contradiction in how the method is described. Section 3 presents the refined label as $\tilde{y_i} = (1-\alpha_i) \cdot s_i + \alpha_i \cdot \tilde{y_i}​$, which appears to be circular since $\tilde{y_i}$ appears on both sides. Algorithm 1 seems to address this by introducing an interpolated label   (Line 10), but then the loss function in Line 15 is defined as $\mathcal{L} = l(f_\theta(x), \tilde{y})$ —using the original noisy label $\tilde{y}$, not the interpolated one. This creates a fundamental problem: if the loss doesn't depend on $\alpha_j$​, how is the gradient $\nabla_{\{\alpha_j\}} \mathcal{L}$ in Line 19 computed, and why wouldn't it just be zero? I've read this section multiple times and cannot reconcile these descriptions. This makes the core mechanism impossible to reproduce or verify.

2. The reliance on Otsu's thresholding feels equally problematic. The entire correction phase hinges on the assumption that $\alpha_i​ values will form a clean bimodal distribution—one mode for clean samples, another for noisy ones. But the paper provides no visualizations or empirical evidence that this actually happens. Otsu's method will always produce a threshold, even if the distribution is unimodal or multimodal in unexpected ways, and in those cases the resulting split would be arbitrary. Why not use more robust clustering methods like Gaussian Mixture Models that can adapt to the actual distribution shape? The choice seems unjustified given how critical this step is to the method's success.

3. My biggest concern, though, is the experimental setup. The paper never compares LINC's final classification performance against other noise correction methods. Table 1 shows noise identification metrics (comparing against AUM and similar methods), but Tables 2 and 3—which contain the main accuracy and AUROC results—only compare "With LINC" versus "Without LINC." This is a simple baseline that tells us the method does something, but not whether it's competitive with existing approaches. The related work mentions standard techniques like bootstrapping and label smoothing, but none of these appear in the experimental comparison. Without this, I can't tell if LINC's multi-stage process offers any real advantage over simpler, well-established methods that would be far easier to implement.
These issues collectively make it hard to recommend acceptance. The methodological ambiguity needs to be resolved before the approach can be reproduced, and the experiments need to demonstrate clear advantages over existing noise correction techniques.

**Questions:**

1. Clarification of the Core Algorithm: I am having difficulty understanding the core mechanism in Algorithm 1.
* Line 10 calculates an interpolated label $\tilde{I}j=(1-\{\alpha{j}\})*s{j}+(\{\alpha{j}\})*\tilde{y}{j}$.
* However, the loss function in Line 15 is defined as $\mathcal{L}=l(f_{\theta}(x),\tilde{y})$, which uses the original, noisy label $\tilde{y}$.
If the loss $\mathcal{L}$ is not a function of $\alpha_j$, how is the gradient $\nabla_{\{\alpha_{j}\}}\mathcal{L}$ in Line 19 computed (i.e., how is it not zero)? Is Line 15 a typo, and should the loss actually be $\mathcal{L}=l(f_{\theta}(x),\tilde{I}_{j})$? This seems critical to the method's function.

2. Justification for Otsu's Method: The correction step in Line 13 hinges on Otsu's method, which, as noted, "exploits bimodality".
* Could the authors please provide a visualization (e.g., a histogram) of the $\alpha_i$ parameter distribution after the warmup phase? This would help validate the core assumption that the values are indeed bimodal.
* What is the method's behavior if this assumption fails and the $\alpha_i$ distribution is not bimodal (e.g., a single broad, unimodal smear)?
* Could the authors justify why Otsu's method was chosen over more robust clustering methods (like a simple Gaussian Mixture Model) that might be less sensitive to this bimodal assumption?

3. Absence of Comparative Baselines in Main Results: The experimental validation in Table 2 (Accuracy) and Table 3 (AUROC)  only compares the model "With LiNC" against "Without LiNC". While Table 1  compares noise identification AUROC, the main tables never compare the final model performance against other noise correction methods.
* Why did the authors not compare LiNC's final accuracy against the established noise-correction techniques (e.g., Bootstrapping , Label Smoothing) discussed in the Related Work? Without this, it is impossible to gauge if this multi-stage method is actually more effective than simpler, standard approaches.

---

### Official Review · Reviewer_XuU7 · 2025-11-04

**Soundness:** 2
**Presentation:** 2
**Contribution:** 2
**Rating:** 2
**Confidence:** 3

**Summary:**

This paper proposes an adaptive label refinement method for addressing the problem of noisy labels in medical learning. The authors introduce a per-sample trust parameter, which is dynamically updated during model training. Experiments on medical datasets demonstrate the superiority of the proposed method. Thanks to its lightweight and adaptive per-sample noise modeling, the approach performs particularly well in scenarios with high label noise (~0.5).

**Strengths:**

1. The idea is clearly presented and easy to understand.
2. The authors conduct extensive experiments on multiple medical datasets for comparison.

**Weaknesses:**

1. The presentation of the results is unclear. For example, in Table 2, the meaning of the arrow symbols is not explained.
2. The title should include the keyword “Medical.” Without it, the title is misleading, as it implies a general adaptive label refinement method rather than one specific to medical data.
3. In the algorithm, $\alpha_j$ in Line 6 is not defined. There are several different $\alpha$ terms, which makes the notation confusing.
4. In the algorithm, Line 18 should read “if epoch > $w$” instead.
5. The authors should provide more discussion regarding the source of supervision.

**Questions:**

1. The proposed method appears somewhat counterintuitive. The authors should further explain why introducing a new trainable parameter can lead to performance improvements.
2. How does the proposed method behave on clean datasets? Since the parameter is adaptive, one would expect it to converge to all 0s or all 1s in the absence of label noise.

---

### Official Review · Reviewer_s9LC · 2025-11-04

**Soundness:** 2
**Presentation:** 2
**Contribution:** 2
**Rating:** 4
**Confidence:** 3

**Summary:**

This paper proposes LiNC, a method for handling label noise in medical imaging by assigning a trainable per-sample trust parameter $\alpha_i$ to each training sample. These parameters interpolate between observed labels and model predictions, with values learned via gradient descent during a warmup phase. After warmup, Otsu thresholding binarizes the α values to determine which labels to correct. Experiments on ten MedMNIST datasets show improvements in accuracy and AUROC.

**Strengths:**

- Simplicity and efficiency: The method adds only one parameter per sample and integrates naturally into existing training workflows without requiring auxiliary models or clean validation sets.
- Interpretability: The learned α values provide direct insight into which samples are likely mislabeled, enabling potential human-in-the-loop auditing.

**Weaknesses:**

- Limited novelty: The core contribution is essentially bootstrapping (Reed et al., 2014) with learnable per-sample mixing weights instead of a fixed global parameter. The paper acknowledges bootstrapping but understates the similarity. The main novelty is making α per-sample and learnable, which is incremental.
- Insufficient baseline comparisons: Tables 2–3 only compare the performance with and without LiNC, without including other noise correction baselines. Moreover, the comparisons lack recent state-of-the-art methods from 2024–2025.
- Degradation on some datasets: OCTMNIST and TissueMNIST show consistent accuracy drops with LiNC (Table 2). The paper mentions these are "hardest tasks" but provides no analysis. Investigate failure modes and provide explanations or guidance on when LiNC should/shouldn't be used.
- The writing style could be further improved, with more attention to consistency in capitalization and punctuation, especially in the section describing the contributions or novelty of the work.

**Questions:**

- Otsu is applied once at epoch $\omega$ (Line 13), making a permanent binary decision. If the model is poorly calibrated at epoch 10, incorrect corrections cannot be reversed. Why this one-shot correction is better than continuous soft interpolation. You can give ablation on key hyperparameters: $\omega$, $\alpha_{lr}$, $\alpha_{wd}$ or some other evidence.
- I am curious about the computational overhead (time, memory) compared to baseline training? How does it scale with dataset size?

---

### Note · Authors · 2025-12-03

**Comment:**

We thank the reviewers for their reviews. We will update our work based on their feedback and address all of their concerns in future work.

**Withdrawal Confirmation:**

I have read and agree with the venue's withdrawal policy on behalf of myself and my co-authors.